# Characterization of Human Immortalized Keratinocyte Cells Infected by Monkeypox Virus

**DOI:** 10.3390/v16081206

**Published:** 2024-07-26

**Authors:** Chaode Gu, Zhiqiang Huang, Yongyang Sun, Shaowen Shi, Xiubo Li, Nan Li, Yang Liu, Zhendong Guo, Ningyi Jin, Zongzheng Zhao, Xiao Li, Hongwei Wang

**Affiliations:** 1State Key Laboratory of Analytical Chemistry for Life Science & Jiangsu Key Laboratory of Molecular Medicine, Medical School, Nanjing University, Nanjing 210093, China; chaode_gu@smail.nju.edu.cn (C.G.); zhiqiang.huang@nju.edu.cn (Z.H.); 2Changchun Veterinary Research Institute, Chinese Academy of Agricultural Sciences, State Key Laboratory of Pathogen and Biosecurity, Key Laboratory of Jilin Province for Zoonosis Prevention and Control, Changchun 130118, China; sean_young@163.com (Y.S.); shi3751058209@163.com (S.S.); lixiubo0507@126.com (X.L.); linan226@126.com (N.L.); l17320675260@163.com (Y.L.); guozd@foxmail.com (Z.G.); ningyik@126.com (N.J.); 3College of Veterinary Medicine, Jilin Provincial Engineering Research Center of Animal Probiotics, Jilin Provincial Key Laboratory of Animal Microecology and Healthy Breeding, Engineering Research Center of Microecological Vaccines (Drugs) for Major Animal Diseases, Ministry of Education, Jilin Agricultural University, 2888 Xincheng Street, Changchun 130118, China; 4College of Veterinary Medicine, Hebei Agricultural University, Baoding 071000, China

**Keywords:** monkeypox virus, HaCaT, TEM, characterization, RNA-seq

## Abstract

Monkeypox virus (MPXV) can induce systemic skin lesions after infection. This research focused on studying MPXV proliferation and the response of keratinocytes. Using transmission electron microscopy (TEM), we visualized different stages of MPXV development in human immortalized keratinocytes (HaCaT). We identified exocytosis of enveloped viruses as the exit mechanism for MPXV in HaCaT cells. Infected keratinocytes showed submicroscopic changes, such as the formation of vesicle-like structures through the recombination of rough endoplasmic reticulum membranes and alterations in mitochondrial morphology. Transcriptome analysis revealed the suppressed genes related to interferon pathway activation and the reduced expression of antimicrobial peptides and chemokines, which may facilitate viral immune evasion. In addition, pathway enrichment analysis highlighted systemic lupus erythematosus pathway activation and the inhibition of the Toll-like receptor signaling and retinol metabolism pathways, providing insights into the mechanisms underlying MPXV-induced skin lesions. This study advances our understanding of MPXV’s interaction with keratinocytes and the complex mechanisms leading to skin lesions.

## 1. Introduction

Monkeypox (mpox), caused by the monkeypox virus (MPXV), is a zoonotic infectious disease. The first human case of monkeypox was reported in 1970 [1]. Since then, human cases of monkeypox have predominantly occurred in central and western Africa. However, beginning in 2022, there was a noticeable increase in monkeypox cases in non-African regions such as the United Kingdom, the European Union, and the United States, sparking considerable interest among researchers [2,3]. In May 2023, the first documented human case of monkeypox was reported in mainland China [4]. MPXV has a broad host range, including humans, monkeys, dogs, and rodents such as marmots, squirrels, rats, and mice [5]. Currently, there is some evidence that MPXV is transmitted by droplets [6], body fluids [7,8], and contaminated materials [9,10,11]. MPXV is relatively stable in the environment [12,13,14], but can be inactivated by alcohol and high temperatures [15,16]. Of particular note is the observation that the 2022 mpox outbreak primarily affected men who have sex with men, possibly due to the increased susceptibility of the rectal mucosa to MPXV [4,17,18,19]. And, those people were mostly co-infected with HIV and treponema pallidum [20,21]. The incubation period for mpox ranges from 5 to 14 days with the upper limit of 21 days [22]. Patients infected with MPXV develop characteristic systemic skin lesions that progress through papules, blisters, pustules, and scabs over a period of 14 to 21 days before the scabs eventually fall off leaving scars [2]. Additionally, the patients usually experience fever, headache, myalgias, and swollen lymph nodes [23].

MPXV belongs to the same Poxviridae family as the smallpox virus, possesses a linear double-stranded DNA genome of approximately 193 kb in size, and encompasses approximately 200 open reading frames [24]. The central regions of the MPXV genome exhibit high levels of conservation and encode proteins associated with crucial events such as viral entry, replication, and maturation. Conversely, the terminal regions of the genome display lower conservation levels and are responsible for encoding proteins involved in host–virus interactions [25]. According to the phylogeny, MPXV strains can be classified into primarily three clades, including the Congo Basin clade (clade I), West Africa clade (clade IIa), and 2022 MPXV (clade IIb) [26]. Clade I is acknowledged to be more virulent, exhibiting an average mortality rate of 10.6%, while that of clade IIa is just 3.6% [27]. In comparison to the aforementioned local MPXV, clade IIb demonstrates a reduced fatality rate (0.46%) [23].

The replication cycle and morphogenesis of poxviruses like vaccinia virus (VACV) have been extensively elucidated [28,29]. The intracellular mature virus (IMV) and extracellular enveloped virus (EEV) represent infectious viral forms that release their viral cores into the cytoplasm through sophisticated mechanisms upon entering into host cells [30]. The poxviruses morphogenesis takes place within specialized compartments known as viral factories (VF) located in the cytoplasm where viral proteins interact with lipids derived from host cells to form crescent structures, an initial visible manifestation of virions [28]. These crescent structures subsequently develop into an immature virus (IV) and their cores further condense into dumbbell-shaped structures, which is a characteristic of IMV. Some IMV particles exit the VF and are transported via the microtubule network in close proximity to the microtubule organizing center, where they are enveloped by a bilayer membrane derived from either the trans-Golgi network (TGN) or the endosome [31]. The resulting virus is referred to as an intracellular enveloped virus (IEV). Upon reaching the cell surface, the outer membrane of the IEV fuses with the plasma membrane, releasing enveloped virions in a process similar to exocytosis. Virions remaining on the cell surface are termed cell-associated enveloped virus (CEV), while those released into the extracellular space are termed EEV [29].

Many cell models, including Vero [32], NHEK [33], and hTERT [34], have been utilized for the characterization of MPXV. The replication cycle and morphogenesis of MPXV in these cells exhibit similarities to those observed in other documented poxviruses [32]. However, certain aspects of MPXV’s viral morphogenesis have not been directly reported, particularly the information regarding the various forms of enveloped viruses. Furthermore, the alterations induced by MPXV in skin cells’ submicrostructure are still poorly understood. HaCaT cells, a human immortalized keratinocyte line commonly used as an in vitro model to study various human skin diseases, have not been used in MPXV research. Incorporating HaCaT cells into MPXV studies could provide valuable insights into how this virus interacts with skin cells and potentially reveal novel aspects of MPXV pathogenesis.

In this study, MPVX derived from the pustular fluid of monkeypox patients was utilized to assess the susceptibility of HaCaT cells towards MPXV. Additionally, we examined the submicroscopic structure and transcriptome alterations in both the virus and host cells following infection using transmission electron microscopy and RNA sequencing.

## 2. Materials and Methods

### 2.1. Cell Culture

Vero E6 cells were obtained from the Changchun Veterinary Research Institute (Changchun, China). Both the Vero E6 cells and immortalized human HaCaT cells (NE Fusenig, Heidelberg, Germany) were cultured in Dulbecco’s Modified Eagle’s Medium (DMEM) supplemented with 10% fetal bovine serum (FBS) and 1% penicillin–streptomycin at 37 °C in a humidified incubator with 5% CO_2_ [35].

### 2.2. Preparation of MPXV

The MPXV strain was provided by the Changchun Veterinary Research Institute, and the MPXV experiments were conducted in a biosafety level III (BSL3) laboratory. Initially, MPXV was successfully isolated from the pustular fluid of patients infected with MPXV in 2023 and was maintained in Vero E6 cells. The culture medium of Vero E6 cells in the logarithmic growth phase was refreshed with DMEM supplemented with 2% FBS and 1% penicillin-streptomycin. Subsequently, the Vero E6 cells were inoculated with MPXV and cultured at 37 °C in a humidified incubator with 5% CO_2_. At 3 days post-infection, the Vero E6 cells infected with MPXV were cryopreserved at a temperature of −80 °C. Finally, MPXV was isolated from the infected Vero E6 cells maintained in DMEM after thawing and collected by centrifugation and used for further studies. The MPXV transferred to HaCaT cells was passaged three times and preserved at −80 °C.

### 2.3. Genome Sequencing and Assembly

The DNA library (Illumina, San Diego, CA, USA) was established using 1 ng DNA extracted from MPXV collected from cell culture medium. The library was then sequenced on an ABI 3730XL instrument (ABI, Foster City, CA, USA). The obtained sequences were assembled using BioEdit 7.2 and were aligned with the reference sequence MPV-M5312_HM12_Rivers (GenBank: NC_063383.1).

### 2.4. MPXV Phylogenetic Analysis

Alignments of nucleotide (nt) sequences with other representatives of known viruses were conducted using MAFFT version 7.471. TrimAI version 1.2 was used to clip ambiguous alignments and a model finder was used to predict the best model. Phylogenies were inferred by IQ-TREE version 1.6.8 with 1000 bootstrap replicates.

### 2.5. DNA Extraction and Quantitative PCR

DNA was extracted from samples in specific MPXV laboratories using the E.Z.N.A. ^®^Viral DNA Kit (Promega, Madison, WI, USA, #D3892-02) following the manufacturer’s specification. The viral DNA copy number of MPXV in the samples was quantified by real-time PCR using GoTaq^®^ qPCR Master Mix (Promega, #A600A). The primer sequences of MPXV-*F3L* were as follows: forward: 5′-CATCTATTATAGCATCAGCATCAGA-3′ and reverse, 5′-GAT ACTCCTCCTCGTTGGTCTAC-3′. Amplification reactions were performed using the ABI7500 system (Roche, Geneva, Switzerland) under 40 cycles of 95 °C for 15 s and 60 °C for 1 min. The viral DNA copy number in the sample was estimated from the measured cycle threshold (Ct) value. Standard curves were fitted using a series of 10-fold dilutions of standard plasmid encoding MPXV-*F3L*. The fitting standard curve was Ct = −3.54X_0_ + 40.71, where X_0_ was the initial viral DNA copy number in the reaction system.

### 2.6. Transmission Electron Microscope

HaCaT cells were infected with 0.5 MOI MPXV. After 24 h, the cells were washed with PBS, scraped, collected, and transferred to a 1.5 mL Eppendorf tube. Centrifugated for 10 min at 1000× *g*, the cells were fixed with 2.5% glutaraldehyde (25% glutaraldehyde diluted by PBS) for 24 h at 4 °C. The specimens were then post fixed in 1% osmium tetroxide for 1 h and washed twice with ddH_2_O after dehydration using a graded series of alcohols from 30% to 100%, increasing progressively with 10% increments. The specimens were then embedded in a mixture of epoxy resin, cut into ultrathin sections using Leica UC6 (Leica, Wetzlar, Germany), and stained with 2% uranyl acetate and lead citrate. The thin sections were observed using a transmission electron microscope (Hitachi7800, Tokyo, Japan) at an acceleration voltage of 120 kV. The Microsoft PowerPoint 2021 measurement tool was utilized to analyze virus and cell structures, considering all image scales.

### 2.7. RNA Sequencing

HaCaT cells were infected with 0.5 MOI MPXV for 24 h. Total RNA was extracted from normal or MPXV-infected HaCaT cells for transcriptome sequencing. The RNA extraction and purification were performed using Freezol reagents (Vazyme, Nanjing, China, #R71102). The quality and purity of each sample were assessed using NanoDrop ND-2000 (Thermo, Norristown, PA, USA). Subsequently, the purified RNA samples were submitted to LC Bio Technology (Hangzhou, China) for library preparation and sequencing on Illumina Novaseq™ 6000 platform. Expression levels of all transcripts were estimated using StringTie 2.2.3 and Ballgown 2.36.0 software, while mRNA expression levels were calculated based on Fragments Per Kilobase of transcript per Million mapped reads (-FPKM) values. Differentially expressed genes (DEGs) with a fold change >2 or <0.5 and *p* < 0.05 were identified using the R package DESeq2. Gene Ontology (GO) enrichment analysis and Kyoto Encyclopedia of Genes and Genomes (KEGG) pathway analysis were then conducted for the differentially expressed mRNAs in both groups.

### 2.8. Statistical Analysis

The statistical significance was assessed using unpaired two-tailed Student’s *t*-test, one-way analysis of variance (ANOVA) followed by Dunnett or Tukey post hoc tests, or two-way ANOVA followed by Tukey post hoc tests. All statistical analyses were conducted using GraphPad Prism 9.0 or R 4.2.3 software. A significance level of *p* < 0.05 was considered statistically significant.

## 3. Results

### 3.1. Phylogenetic Analysis of MPVX Strains

Genome sequencing and maximum likelihood estimation were employed for the phylogenetic analysis of the MPVX strain (hMpxV/China/GZ8H-01/2023) derived from the pustular fluid of mpox patients infected with MPXV in 2023. The result demonstrated that the hMpxV/China/GZ8H-01/2023 isolate belongs to MPXV clade IIb, which strongly shares its homology with the MPXV strains circulating in the United States in 2022 (Figure 1).

### 3.2. Growth Kinetics

We investigated the infectivity of MPXV in HaCaT cells. HaCaT cells were infected with 0.5 MOI of MPXV, and subsequent cytopathic changes were observed over time using microscopy. At 24 h post-infection, the HaCaT cells exhibited cytopathic effects (CPE) and formed plaques in the adherent cell monolayer (Figure 2A). Initially, the infected cells displayed elongated antennae, and gradually became rounded prior to detaching from the culture vessel wall as the infection progressed (Figure 2B). When different initial concentrations of MPXV (0.1 MOI and 0.5 MOI) were used to infect HaCaT cells, the viral copy numbers in the cell culture supernatant increased over time and reached a plateau with a concentration of approximately 1 × 10^7^/mL at 72 h post-infection (Figure 2C). At 24 and 48 h post-infection, the proliferation rate of MPXV in HaCaT cells was significantly higher than that in Vero-E6 cells (Figure 2D), indicating efficient proliferation of MPXV in HaCaT cells.

### 3.3. Ultrastructural Analysis of MPXV in HaCaT Cells

Based on the existing investigations and hypotheses regarding VACV, we present a description of MPXV morphogenesis in HaCaT cells using transmission electron microscopy images. The images showed that MPXV exclusively proliferated in the cytoplasm of HaCaT cells (Figure 3A). In addition, the two primary forms of mature virions, unencapsulated IMV and encapsulated IEV, both exhibited a typical dumbbell-shaped viral core (Figure 3B,C). We then further investigated the morphogenesis of MPXV after HaCaT cells had been infected. The process from the emergence of the crescent structure to the formation of IV predominantly occurred within the VF which is characterized by a high electron density, granular appearance, and well-defined edges (Figure 3D). Figure 3E illustrated the morphological changes in MPXV from the viral crescent to the mature virus stage. Subsequently, the mature virions were enveloped by Golgi TGN or endosomes and underwent a transformation into an IEV (Figure 3F). Notably, the IEV possessed two additional membranes compared to IMV and were transported towards the cell periphery after formation (Figure 3G). Upon approaching the cell membrane, the outer membrane of the IEV fused with the plasma membrane and subsequently exited the cell by exocytosis (Figure 3H). These extracellular virus particles were enclosed by singular membrane, and the virions in this state can either be retained near the plasma membrane as CEV (Figure 3I) or released as an EEV (Figure 3J). Both the CEV and EEV possessed one additional membrane compared to the IMVs.

### 3.4. Ultrastructural Analysis of HaCaT Cells Infected with MPXV

After infection with MPXV, significant morphological changes in the mitochondria and endoplasmic reticulum (ER) were observed. During the early stage of MPXV infection, a large number of virions were attached to the cell membrane surface, while only a few were present in the cytoplasm (Figure 4A). At this stage, the mitochondria exhibited a tubular or elliptical shape with clearly visible internal ridge structures (Figure 4A). As the MPXV infection progressed, the mitochondria became aberrant and displayed a spherical morphology with diameters ranging from 0.57 μm to 1.70 μm, similar to normal mitochondria (Figure 4B). These spherical mitochondria retained their inner membrane, outer membrane, and matrix, whereas their ridge structures were reduced or even disappeared, and the matrices were filled with granular contents (Figure 4B). Additionally, numerous IVs were often found surrounding the spherical mitochondria, which appeared to be associated with viral morphogenesis (Figure 4B). Figure 4C showed different morphologies of spherical mitochondria with different degrees of membrane damage. Meanwhile, the normal reticular structure of rough ER (rER) was absent, replaced by vesicles and ribosome-bound fragments (Figure 4D). Figure 4E exhibits the moment when the rER, adjacent to the nucleus, extended outward and protruded to form vesicles. Furthermore, the presence of cabbage-like multimembrane vesicle structures was observed in MPXV-infected HaCaT cells, which typically appeared in the late stages of infection alongside spherical mitochondria and vesicles derived from the rER (Figure 4F). Interestingly, certain HaCaT cells exhibited a C-type nucleus with an opening directed toward the regions of MPXV concentration (Figure 4G). These findings suggest that MPXV infection is capable of inducing various structural abnormalities in HaCaT cells, including mitochondrial spheroidization, vesiculation of rER, generation of multimembrane vesicles, and nuclear invagination, revealing the cellular damage caused by viral infection.

### 3.5. Transcriptomic Analysis of HaCaT Cells Infected with MPXV

To further elucidate the alterations in gene transcription levels of HaCaT cells following MPXV infection, we conducted a transcriptome analysis on HaCaT cells at 24 h post-infection. A substantial number of viral transcripts were detected within the cells, and these transcripts were aligned to the reference genome of MPXV. Notably, viral genes *OPG069*, *OPG138*, *OPG070*, *OPG139*, and *OPG137* exhibited the highest transcription levels and primarily encoded viral core proteins and membrane proteins (Figure 5A). After MPXV infection, a total of 2302 genes in HaCaT cells displayed significant changes, among which 1247 genes were up-regulated while 1055 genes were down-regulated (Log2 (fold change) > 1, adjusted *p* < 0.05) (Figure 5B). Heat maps depicting differentially expressed genes’ (DEGs) expression revealed alterations in certain genes associated with the interferon pathway and IL-17 signaling pathway (Figure 5C). Moreover, the expression levels of specific inflammation-related transcription factors, such as *JUN* and *FOS,* were up-regulated, whereas that of their downstream interferon pathway genes like *GBP2* and *STAT2* were down-regulated. Keratinocytes typically express antimicrobial peptides (*S100A7/8/9*) or chemokines (*CCL2*) to recruit immune cells for defense against external pathogens. However, the mRNA levels of these factors were suppressed upon MPXV infection, which may be a strategy employed by MPXV to elude immune surveillance. KEGG analysis demonstrated the involvement of DEGs in multiple signaling pathways including the systemic lupus erythematosus pathway (Figure 5D). In addition, the GO enrichment analysis indicated that DEGs were involved in the biological processes such as type I interferon response (Figure 5E). Gene set enrichment analysis (GSEA) revealed a remarkable upregulation of nucleosome assembly related genes, suggesting that MPXV infection may affect the chromatin structure of host cells. In addition, GSEA showed that the systemic lupus erythematosus pathway was activated, while the Toll-like receptor pathway and retinoid metabolism were inhibited. The results suggest a correlation between the proliferation of MPXV and the inhibition of host immune pathways. At the same time, MPXV may disrupt retinoid metabolism to induce skin lesions (Figure 5F).

## 4. Discussion

The global monkeypox outbreak in 2022 has prompted researchers to intensify their focus MPXV. MPXV primarily infects human skin, causing lesions in which the researchers were able to isolate the virus [36,37]. Previous reports have demonstrated that MPXV is capable of infecting skin organoids [38] and NHEK cells [33], confirming the possibility of skin cell infection with MPXV. However, a comprehensive characterization of keratinocytes infected with MPXV remains incomplete. In this study, we investigated the growth dynamics of MPXV proliferation in in vitro infected HaCaT cells. Additionally, transmission electron microscopy was employed to visualize the morphogenesis of the virus and submicroscopic structural changes in HaCaT cells following viral infection. Meanwhile, RNA sequencing was utilized to analyze transcriptomic changes in both host and viral genes within the MPXV-infected HaCaT cells.

The sequence comparison and phylogenetic tree analysis indicated that the MPXV strain belonged to clade Ⅱb. Subsequently, MPXV was utilized to infect HaCaT cells. At 24 h post-infection, CPE manifested in the cells, plaque formation occurred in the monolayer cells, and filopodia-like projections emerged. Recently, a study revealed that trophoblast cells infected with MPXV also exhibit filopodia-like projections, suggesting the potential involvement of this structure in intercellular transmission of the virus [39]. As the infection progressed, the cells transited to a rounded morphology and aggregated together, and ultimately, their ability to adhere was impaired. Our data indicated that the number of viral copies increased over time before 72 h post-infection and subsequently reached a plateau, which was consistent with the typical growth curve of orthopoxvirus.

Furthermore, the transmission electron microscopy results exhibited different stages of MPXV morphogenesis in HaCaT cells, including viral crescent structures, IVs, and IMVs. Notably, the viral crescent structures and IVs were primarily generated within the specialized structures known as VF that characterized by high electron density, granular appearance, and well-defined boundaries. These particles with high electron density likely consisted of the proteins and nucleic acids essential for viral synthesis.

The various morphological virions of mature poxviruses play a pivotal role in determining the infectivity of viruses. MPXV also possesses four different types of virions, including IMV, IEV, CEV, and EEV. However, the characterization of the four types of MPXV remains incomplete. An IMV requires the two additional membrane layers from the Golgi TGN or endosome, resulting in the formation of an IEV that is without infectivity. The IEV further transports to the cell membrane via microtubules [40], where its outermost membrane fuses with the cell membrane and is then conveyed to the cell surface through exocytosis. The enveloped virus that remains on the cell surface becomes a CEV, while the released enveloped virus transforms into an EEV. The CEV plays a crucial role in viral cell-to-cell transmission by attaching to the plasma membrane and inducing actin tail formation, which propels CEV towards neighboring cells. Conversely, the EEV facilitates the long-distance transmission of the virus in vivo or in vitro [41]. Our transmission electron microscopy results provided morphological observations of MPXV enveloped viruses during the late infection stage, indicating the consistent morphogenesis between MPXV and other orthopoxviruses [29].

In previous reports, the sub-microstructure of keratinocytes infected with MPXV has not been adequately described. We observed that post viral infection, a portion of HaCaT cells showed a C-type nucleus with an opening directed towards the region concentrated with the virus. This phenomenon may be attributed to the extensive formation of independent vesicular structures derived from rER during MPXV replication, resulting in the reduction in rER adjacent to the nuclear membrane. These ribosome-associated vesicles likely facilitate viral protein synthesis more efficiently. Some investigations have established that the morphology of mitochondria changes when stimulating by external stressors like hypoxia [42]. Correspondingly, significant mitochondrial alterations were observed in the HaCaT cells during the later stages of infection. After virus invasion, the mitochondria underwent expansion and transformed into a rounded shape, while losing their internal ridge structure and emerging granular contents, which were presumed to consist of viral proteins and nucleic acids synthesized by using the semi-autonomy of mitochondria. Subsequently, the outer membrane of the spherical mitochondria disintegrated, which appeared to provide additional resources for viral proliferation. Notably, abundant IV often existed near the mutated spherical mitochondria.

The transcriptome analysis revealed significant alterations in gene transcription levels of HaCaT cells infected with MPXV. Following microbial infection, crucial immune transcription factors like NF-κB and interferon regulatory factor IRF are stimulated by the mode receptor PRR signal pathway, which involves numerous host cofactors such as MYD88, TRAM, TIRAP, and TRIF [43,44]. Simultaneously, keratinocytes secrete antimicrobial peptides, IFN, chemokines, and other immune mediators to recruit immune cells to combat external pathogens. Therefore, we focused on the changes in these genes after MPXV infection. The data demonstrated a manifest reduction in the transcriptional levels of MYD88, IRF1/2/6 and STAT1/2 genes involved in interferon regulation. Meanwhile, the levels of antimicrobial peptide S100A7/8/9 and chemokine CCL2/CCL17/CXCL10 were also down-regulated, which may be a potential mechanism for MPXV immune evasion. Consistent with the reports regarding NHEK cells, the GESA results indicated that the genes transcriptional levels associated with the nucleosome assembly pathway were significantly up-regulated [33]. The maintenance of retinoid metabolism homeostasis plays a crucial role in regulating skin growth and development. Interestingly, the dates showed that the retinoid metabolism pathway was suppressed in HaCaT cells upon MPXV infection, which could potentially result in virus-induced skin lesions.

## 5. Conclusions

In this study, HaCaT cells were employed to assess the impact of MPXV infection. The graphs of TME illustrated the transformation of MPXV from viral crescent structures to an IMV and the formation of the enveloped virus, as well as the procession of viral transportation via exocytosis. Furthermore, the results revealed submicroscopic structural alterations in the host cells. The utilization of in vitro skin cell models may promote the development of novel therapeutic agents targeting MPXV while also facilitating further investigations into the mechanisms underlying MPXV-induced cutaneous lesions.

## Figures and Tables

**Figure 1 viruses-16-01206-f001:**
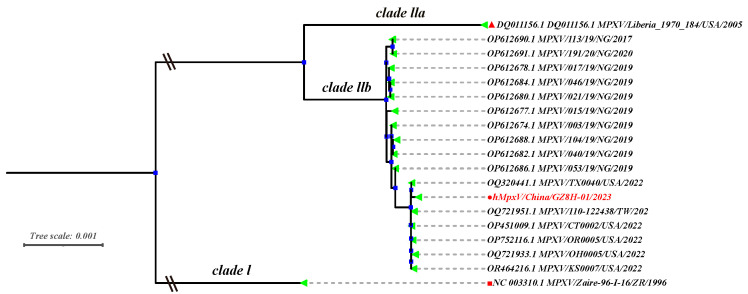
The phylogenetic analysis of hMpxV/China/GZ8H-01/2023. Phylogenetic analysis showed the hMpxV/China/GZ8H-01/2023 isolate belonged to clade IIb of MPXV, highlighted in red.

**Figure 2 viruses-16-01206-f002:**
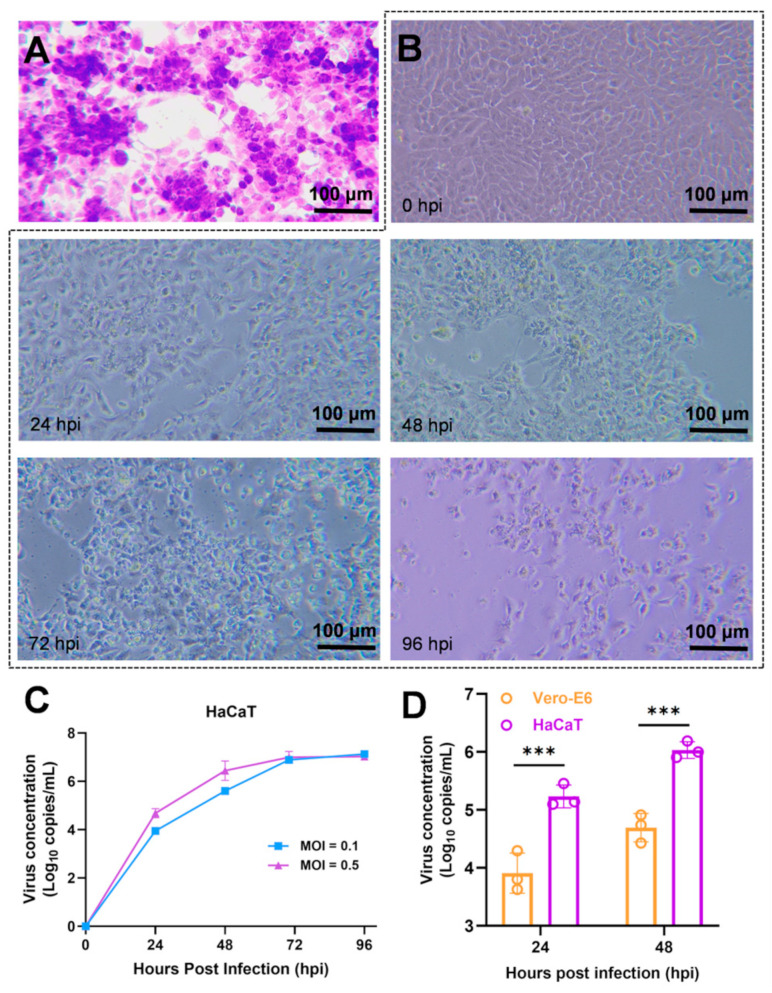
The growth dynamics of MPXV in HaCaT cells. (**A**) Cytopathic effects (CPE) of HaCaT cells infected with MPXV by crystal violet staining. (**B**) CPE of HaCaT cells infected with 0.5 MOI MPXV at different time points (0, 24, 48, 72, and 96 hpi). (**C**) HaCaT cells infected with 0.1 or 0.5 MOI MPXV, virus concentration in the supernatant were detected and recorded at 0, 24, 48, 72, and 96 hpi, respectively (*n* = 3). (**D**) Infection of Vero-E6 and HaCaT cells with 0.5 MOI MPXV, followed by virus concentration in the supernatant detection and recording at 24 and 48 hpi, respectively (*n* = 3). Data were analyzed by one-way ANOVA and *t*-test. *p*  <  0.05 was considered statistically significant. “***” represents *p*  <  0.001.

**Figure 3 viruses-16-01206-f003:**
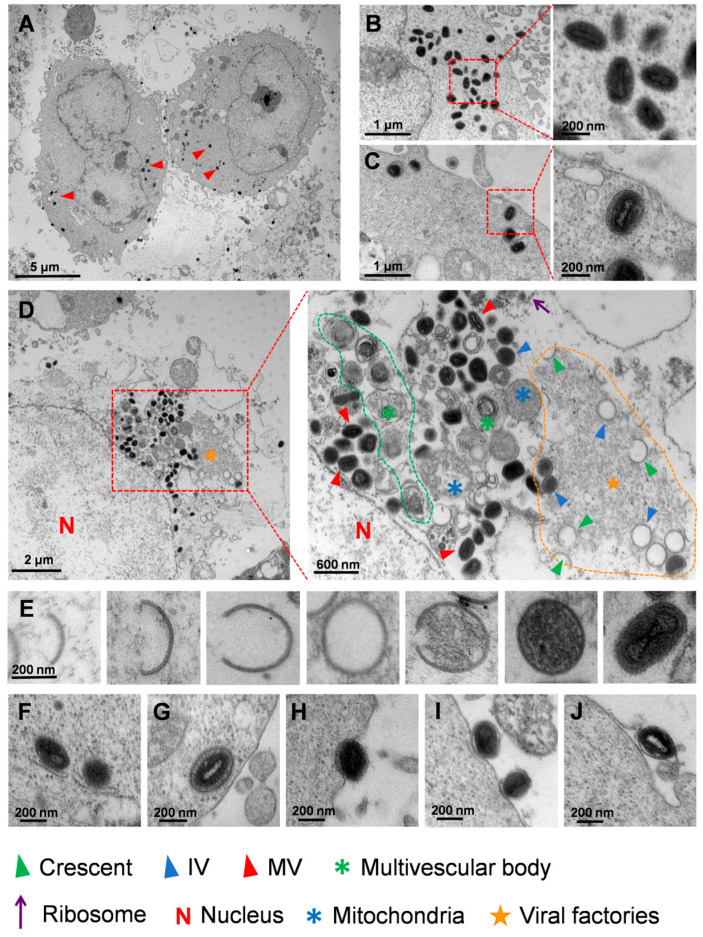
Morphogenesis of MPXV in HaCaT cells. (**A**) Numerous MPXV virions were observed in cytoplasmic matrix of the infected HaCaT cells. (**B**,**C**) The mature MPXV virions presented two different forms, the unencapsulated virion IMV and the encapsulated virion IEV. (**D**) The viral factory (marked with a yellow pentagram) encompassed various structures, including crescent structure (green triangle), immature (blue triangle) and mature (red triangle) viral particles, as well as multi-membrane vesicles (green asterisk), and abnormal mitochondria (blue asterisk). (**E**) Transformation of MPXV from viral crescent structures to IMV. (**F**) The mature virions were enveloped by membranes. (**G**) IEV at the periphery of the cell. (**H**) IEV was transported out of the cell through exocytosis. (**I**) The extracellular virions enveloped by singular membrane near the plasma membrane were termed as CEV. (**J**) Released extracellular virions were termed as EEV.

**Figure 4 viruses-16-01206-f004:**
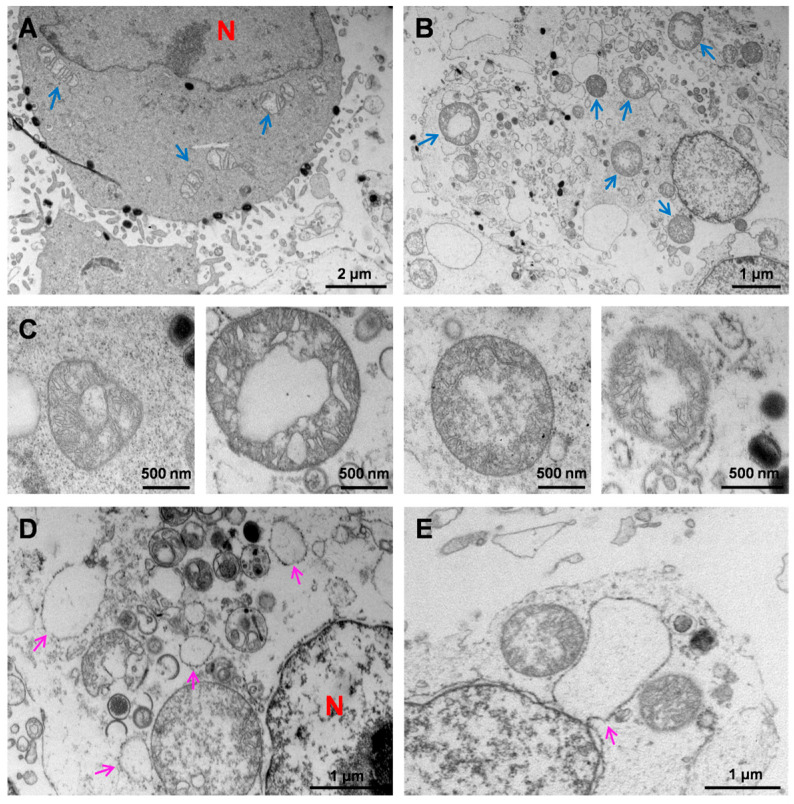
Ultrastructural features of HaCaT cells infected with MPXV. (**A**) The HaCaT cells at the early stage of MPXV infection exhibited a substantial presence of EEV attached to the cell membrane surface, along with a few virions within the envelope, while maintaining an intact mitochondrial structure. (**B**) Mitochondria in late-stage infected cells showed an enlarged and rounded morphology, accompanied by a reduction in or the disappearance of cristae, with the matrices filled with granular contents. (**C**) Different forms of spherical mitochondria showed varying degrees of damage to their inner and outer membranes or cristae. (**D**) Numerous vesicle structures derived from rER were observed in the infected HaCaT cells. (**E**) The moment when the rER membrane protruded outward to form a vesicle. (**F**) At the late stage of MPXV infection, abundant multi-membrane vesicle structures appeared in HaCaT cells, along with the presence of mature virions nearby. (**G**) The nucleus in a portion of infected HaCaT cells exhibited a C-type morphology, with an opening facing the region of MPXV aggregation.

**Figure 5 viruses-16-01206-f005:**
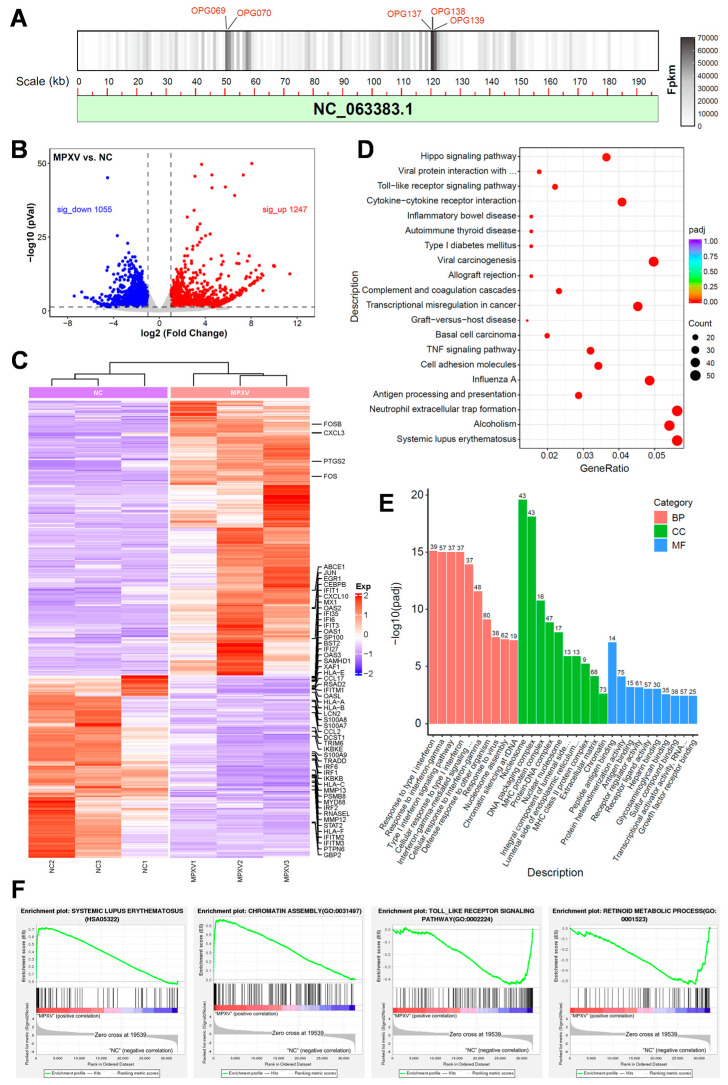
RNA-seq analyses of the MPXV-infected HaCaT cells. (**A**) The location of MPXV transcript in viral genome. (**B**) The volcano plot of DEGs regarding uninfected and 24 h post-infected HaCaT cells (Log2 (fold change) > 1, adjusted *p* < 0.05). The red dots represent upregulated genes while the blue dots represent downregulated genes. (**C**) The heat map of DEGs, in which the genes associated with interferon and IL17 signaling pathways were marked. (**D**) KEGG analysis of the DEGs. (**E**) DEGs were categorized based on their gene ontology (GO) terms related to biological process (BP), cell component (CC), and molecular function (MF). (**F**) Gene set enrichment analysis of specified gene sets, including lupus erythematosus signaling pathway, body assembly, Toll-like receptor signaling pathway, and retinoid metabolism.

## Data Availability

All data generated and analyzed in this study are included in this article.

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
