# Peer review of "Characterization of Human Immortalized Keratinocyte Cells Infected by Monkeypox Virus"

_viruses, 2024, doi:10.3390/v16081206_

Round 1

Reviewer 1 Report

Comments and Suggestions for Authors

This is an article with robust materials and methods. The quality of the images is very good, and I find it a very interesting article.

Perhaps the only drawback is that the current presence and relevance of mpox is minor, but that does not diminish the work of the authors and I must congratulate them and recommend its publication.

Author Response

Comments for the Authors (Reviewer #1)

This is an article with robust materials and methods. The quality of the images is very good, and I find it a very interesting article.

Perhaps the only drawback is that the current presence and relevance of mpox is minor, but that does not diminish the work of the authors and I must congratulate them and recommend its publication.

Reply: We appreciate the supportive comments from reviewer #1. Given the current lack of available vaccines to prevent mpox, a basic understanding of morphogenesis is necessary for MPXV research. It is essential to understand the pathogenic mechanisms of MPXV in vitro cells to characterize its infection process. Such understanding will provide fundamental knowledge to intervene in viral replication and control the spread of the virus.

Reviewer 2 Report

Comments and Suggestions for Authors

The manuscript entitled: “Characterization of HaCaT cells infected by monkeypox virus”, focuses on assessing the MPXV interference with keratinocyte functions during an in vitro infection. The obtained data are thorough and well-presented. The Authors analyzed the replication cycle of poxvirus in cells to which it has affinity during natural infection and obtained very interesting results that will lead to a better understanding of viral morphogenesis. Additionally, the Authors presented how MPXV affects the gene expression in keratinocytes, indicating possible pathways of viral evasion. These data allow for better recognition of the relationship between the target cells and MPXV and also for elucidation of the immunobiology of orthopoxviruses.

However, some errors in the article should be corrected before publication.

 The Authors should change the name of the disease caused by MPXV from “human monkeypox” to “mpox” as is recommended by the WHO.

 Line 42: The Authors should change the phrase “monkeypox is relatively stable …” to “MPXV is relatively stable …” since the virus, not the disease, is stable in the environment.

 Line 44: the outbreak concerns a disease, therefore, please change “…that the 2022 MPXV outbreak...” to “...that the 2022 mpox outbreak...”

 Line 47: HIV is certainly not a disease, please rephrase the sentence.

 Lines 47-48: please, rephrase the sentence from “The incubation period for MPXV ranges from 5 to 21 days” to the more accurate “The incubation period for mpox ranges from 5 to 14 days with the upper limit of 21 days”, since incubation period applies to mpox, not to the virus.

 Lines 64-79: the figure showing virus replication should be placed together with a description of the replication cycle; this will make it easier to understand both, this mechanism and the concept of the manuscript.

 Section 2.2

Line 104: the information regarding MPXV, presented in lines 314-315, should be placed in this section. Moreover, it should be clearly defined how many times MPXV was passaged on the Vero E6 cell line before the virus was transferred to HaCaT cells.

 Line 110: the description of cryopreservation should be included.

 Section 2.5

Line 124: the specification of the clinical samples as the source of the field MPXV as well as the number of analyzed samples should be included. Moreover, it should be stated if one patient equaled one sample or if one patient was the source of more than one sample.

 Section 2.6

Line 137: the infection protocol indicates an infection dose as PFU/ml; it has to be changed into the multiplicity of infection (MOI) since it is critically important how many keratinocytes were infected with the defined dose of MPX virus.

Line 138: Please, include the conditions of collecting cells, and washing; also please, provide the information regarding the presence of the viral particles in the supernatant, were they present or not.

 Section 2.7

The Authors should specify whether the infection protocol for RNA sequencing was the same as for TEM analysis; if not, the description of the new protocol should be included.

 Section 3.2

Line 178: it is widely accepted to use MOI instead of PFU.

 Section 3.5

The quality of the Fig 5 is not acceptable. It has to be improved. In this form is not possible to follow presented data. The description of individual graphs and analyzed genes is illegible.

 Discussion:

Lines 303-304: the sentence should be rewritten, since in its current form it is difficult to follow the main thought.

Line 323: should be “data” not “dates”,

Line 335: change “form” to “from”.

Lines 347-362: the presented results are very interesting and communicative, as they show new paths to understanding the morphogenesis of MPXV, however, they should be discussed in the context of the course of cell culture infection with other poxviruses since the links between mitochondria and viral particles formation have been proved in that model.

Line 367: the word “keratinocytes” should start with a small letter.

Line 371: should be: ”…genes that are involved…”.

 I recommend this manuscript for publication if corrected accordingly.

Comments on the Quality of English Language

Minor editing of English language is required

Author Response

Point-by-point response to the comments of Reviewer #2 

Comments for the Authors (Reviewer #2)

The manuscript entitled: “Characterization of HaCaT cells infected by monkeypox virus”, focuses on assessing the MPXV interference with keratinocyte functions during an in vitro infection. The obtained data are thorough and well-presented. The Authors analyzed the replication cycle of poxvirus in cells to which it has affinity during natural infection and obtained very interesting results that will lead to a better understanding of viral morphogenesis. Additionally, the Authors presented how MPXV affects the gene expression in keratinocytes, indicating possible pathways of viral evasion. These data allow for better recognition of the relationship between the target cells and MPXV and also for elucidation of the immunobiology of orthopoxviruses.

However, some errors in the article should be corrected before publication.

General response to the reviewer #2 comments:

We greatly appreciate your time and effort in reviewing our manuscript entitled "Characterization of HaCaT cells infected by monkeypox virus" (Manuscript ID: viruses-3066767) and providing us with your valuable feedback. Your comments have been thoroughly reviewed and we have made the necessary revisions to improve the quality of our manuscript. The revised sections are highlighted in yellow in the updated manuscript. Below, we outline the major corrections made in response to the reviewer comments:

  • The Authors should change the name of the disease caused by MPXV from “human monkeypox” to “mpox” as is recommended by the WHO.

Reply: Thank you for your valuable suggestions. Since this issue is also related to the following questions, including (1), (2), (3), (4), (5), (7), (16), (17), (19), and (20), we have corrected the errors in the article accordingly.

  • Line 42: The Authors should change the phrase “monkeypox is relatively stable …” to “MPXV is relatively stable …” since the virus, not the disease, is stable in the environment.

Reply: Thank you for your recommendation. We have incorporated the suggested change into line 42 of the manuscript.

  • Line 44: the outbreak concerns a disease, therefore, please change “…that the 2022 MPXV outbreak...” to “...that the 2022 mpox outbreak...”

Reply: Thank you for your recommendation. We have incorporated your changes and made the appropriate changes in line 44 of the manuscript.

  • Line 47: HIV is certainly not a disease, please rephrase the sentence.

Reply: Thank you for your recommendation. We have addressed this by revising the sentence to "And those individuals were largely co-infected with HIV and treponema pallidum" in lines 46-47 of the manuscript.

  • Lines 47-48: please, rephrase the sentence from “The incubation period for MPXV ranges from 5 to 21 days” to the more accurate “The incubation period for mpox ranges from 5 to 14 days with the upper limit of 21 days”, since incubation period applies to mpox, not to the virus.

Reply: We have taken your amendments and made the corresponding changes in the manuscript in lines 47-48.

  • Lines 64-79: the figure showing virus replication should be placed together with a description of the replication cycle; this will make it easier to understand both, this mechanism and the concept of the manuscript.

Reply: Thanks for your insightful feedback. We appreciate the suggestion to include the figure depicting virus replication along with the description of the replication cycle to enhance understanding. However, we have decided not to include the mechanism diagram in this section because it is uncommon to include figures in the Introduction. The mechanism we describe is thoroughly illustrated with diagrams in the referenced literature, which we believe provides sufficient clarity for the reader.

  • Line 104: the information regarding MPXV, presented in lines 314-315, should be placed in this section. Moreover, it should be clearly defined how many times MPXV was passaged on the Vero E6 cell line before the virus was transferred to HaCaT cells.

Reply: We have taken your amendments and made the corresponding changes in the manuscript in lines 106-108. The MPXV transferred to HaCaT cells was passaged third times.

  • Line 110: the description of cryopreservation should be included.

Reply: We added the relevant information “The MPXV transferred to HaCaT cells was passaged third times and preserved at -80°C.” in the manuscript in lines 114-115.

  • Line 124: the specification of the clinical samples as the source of the field MPXV as well as the number of analyzed samples should be included. Moreover, it should be stated if one patient equaled one sample or if one patient was the source of more than one sample.

Reply:

We need to clarify that our work only involves isolation and in vitro infection and does not involve clinical samples. We have already updated the methodological statement accordingly.

For the part of the experiment "DNA Extraction and Quantitative PCR", we measured the virus concentration in the supernatant of cell culture instead of clinical samples using q-PCR. We have added the relevant description in the manuscript in lines 188, 197, and 199.

  • Line 137: the infection protocol indicates an infection dose as PFU/ml; it has to be changed into the multiplicity of infection (MOI) since it is critically important how many keratinocytes were infected with the defined dose of MPX virus.

Reply: We apologize for the oversight. The term "PFU/mL" in the article should actually be "PFU/cell" as correctly stated in the results section. We have now replaced it with "MOI" to accurately reflect the infection protocol and provide clarity regarding the number of keratinocytes infected with the defined dose of MPXV.

  • Line 138: Please, include the conditions of collecting cells, and washing; also please, provide the information regarding the presence of the viral particles in the supernatant, were they present or not.

Reply: Thank you for your suggestion. We detected the virus particles in the supernatant, as shown in the corresponding figure 2C. In addition, we have described this in the corresponding Methods section.

We have added the following to the description of the relevant experimental conditions: "HaCaT cells were infected with 0.5 MOI MPXV. After 24 hours, the cells were washed with PBS, scraped, collected, and transferred to a 1.5 mL Eppendorf tube. The samples were then centrifuged at 1000g for 10 minutes." In addition, Figure 2C shows the virus concentration in the supernatant of keratinocytes infected with MPXV. This should provide a full understanding of the experimental conditions and the presence of virus particles in the supernatant.

  • The Authors should specify whether the infection protocol for RNA sequencing was the same as for TEM analysis; if not, the description of the new protocol should be included.

Reply: Yes, the infection protocol for RNA sequencing and TEM analysis was the same. HaCaT cells were infected with 0.5 MOI MPXV, and cells were collected 24 hours later for follow-up experiments. We made corresponding changes in the manuscript. You will find the description in the manuscript in lines 151.

  • Line 178: it is widely accepted to use MOI instead of PFU.

Reply: We have taken your amendments and made the corresponding changes in the manuscript in line 182.

  • The quality of the Fig 5 is not acceptable. It has to be improved. In this form is not possible to follow presented data. The description of individual graphs and analyzed genes is illegible.

Reply: We have updated the image in Figure 5 with a higher resolution one.

  • Lines 303-304: the sentence should be rewritten, since in its current form it is difficult to follow the main thought.

Reply: We revised this sentence to “MPXV primarily infects human skin, causing lesions in which the researchers were able to isolate the virus” in the manuscript in lines 309-310. This sentence mainly emphasizes that MPXV can cause skin lesions and can be isolated from skin lesions to introduce subsequent studies on monkeypox in skin.

  • Line 323: should be “data” not “dates”.

Reply: We have taken your amendments and made the corresponding changes in the manuscript in line 326.

  • Line 335: change “form” to “from”.

Reply: We have taken your amendments and made the corresponding changes in the manuscript in line 338.

  • Lines 347-362: the presented results are very interesting and communicative, as they show new paths to understanding the morphogenesis of MPXV, however, they should be discussed in the context of the course of cell culture infection with other poxviruses since the links between mitochondria and viral particles formation have been proved in that model.

Reply: Thank you for your suggestion. Cells may exhibit specialized submicroscopic structures after stimulation. However, there are few reports about the change of this submicroscopic structure. Under hypoxic condition, mitochondria will phagocytose lysosomes and form globules with reduced ridge structure. We observed similar changes in mitochondria in keratinocytes infected with monkeypox virus. However, the literature currently lacks comprehensive insights into other potential submicroscopic alterations that may occur during viral infection. Our interpretations are based on informed speculation within the existing knowledge framework.

  • Line 367: the word “keratinocytes” should start with a small letter.

Reply: We have taken your amendments and made the corresponding changes in the manuscript in line 370.

  • Line 371: should be: ”…genes that are involved…”.

Reply: We have taken your amendments and made the corresponding changes in the manuscript in line 374.

Finally, we would like to express our great appreciation to you again for your professional review work on our manuscript.
